# Amortized Variational Inference: When and Why?

**Charles C. Margossian**[1]                                    **David M. Blei**[2]

[1]Center for Computational Mathematics , Flatiron Institute , New York, New York, USA
[2]Department of Computer Science and Statistics, Columbia University, New York, New York, USA

## Abstract

In a probabilistic latent variable model, factorized (or mean-field) variational inference (F-VI) fits a separate parametric distribution for each latent variable. Amortized variational inference (A-VI) instead learns a common inference function, which maps each observation to its corresponding latent variable's approximate posterior. Typically, A-VI is used as a step in the training of variational autoencoders, however it stands to reason that A-VI could also be used as a general alternative to F-VI. In this paper we study when and why A-VI can be used for approximate Bayesian inference. We derive conditions on a latent variable model which are necessary, sufficient, and verifiable under which A-VI can attain F-VI's optimal solution, thereby closing the *amortization gap*. We prove these conditions are uniquely verified by simple hierarchical models, a broad class that encompasses many models in machine learning. We then show, on a broader class of models, how to expand the domain of AVI's inference function to improve its solution, and we provide examples, e.g. hidden Markov models, where the amortization gap cannot be closed.

## 1 INTRODUCTION

A latent variable model is a probabilistic model of observations $\mathbf{x} = x_{1:N}$ with corresponding local latent variables $\mathbf{z} = z_{1:N}$ and global latent parameters $\theta$. With a model $p(\theta, \mathbf{z}, \mathbf{x})$ and an observed dataset $\mathbf{x}$, the central computational problem is to approximate the posterior distribution of the latent variables $p(\theta, \mathbf{z} \mid \mathbf{x})$.

One widely-used method for approximate posterior inference is *variational inference* (VI) [Jordan et al., 1999, Blei et al., 2017]. VI sets a parameterized family of distributions

$\mathcal{Q}$ and finds the member of the family that minimizes the Kullback-Leibler (KL) divergence

$$q^* = \arg\min_{q \in \mathcal{Q}} \mathrm{KL}\left(q(\theta, \mathbf{z}) \,||\, p(\theta, \mathbf{z} \mid \mathbf{x})\right). \qquad (1)$$

VI then approximates the posterior with the optimized $q^*$. (In practice, VI finds a local optimum of Eq. 1.)

To fully specify the VI objective of Eq. 1, we must decide on the variational family $\mathcal{Q}$ over which to optimize. Many applications of VI use the *fully factorized family*, also known as the *mean-field family*. It is the set of distributions where each variable is independent,

$$\mathcal{Q}_{\mathrm{F}} = \left\{ q : q(\theta, \mathbf{z}) = q_0(\theta) \prod_{n=1}^{N} q_n(z_n) \right\}, \qquad (2)$$

and where the notation $q_n$ clarifies that there is a separate factor for each latent variable. The factorized family underpins many applications where fast computation is desired to fit high-dimensional models to large data sets [e.g Bishop et al., 2002, Blei, 2012, Giordano et al., 2023]. We call an algorithm that optimizes Eq. 1 over $\mathcal{Q}_{\mathrm{F}}$ *factorized variational inference* (F-VI).

While the factorized family involves a separate factor $q_n(z_n)$ for each latent variable, recent applications of VI have explored the *amortized variational family* [e.g Kingma and Welling, 2014, Tomczak, 2022]. In this family, the latent variables are again independent. But now the variational distribution of $z_n$ is governed by an *inference function* $f_\phi(x_n)$,

$$\mathcal{Q}_{\mathrm{A}} = \left\{ q : q(\theta, \mathbf{z}) = q_0(\theta) \prod_{n=1}^{N} q(z_n \,;\, f_\phi(x_n)) \right\}. \qquad (3)$$

The inference function $f_\phi$ maps each $x_n$ to the parameters of its corresponding latent variable's approximating factor $q_n(z_n)$. Optimizing Eq. 1 now amounts to fitting an approximate posterior $q_0(\theta)$ and the inference function $f_\phi$. Such an algorithm is called *amortized variational inference* (A-VI).

The canonical application of amortization is in the *variational autoencoder* (VAE) [Kingma and Welling, 2014, Rezende et al., 2014], where A-VI is used to do variational

*Accepted for the 40th Conference on Uncertainty in Artificial Intelligence* (UAI 2024).

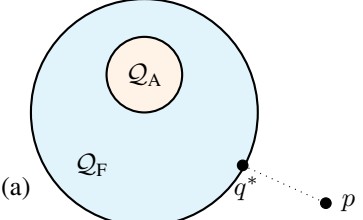 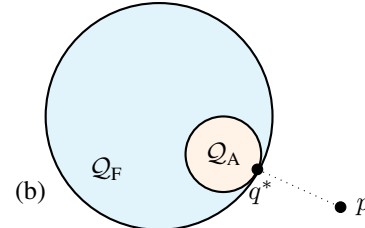

Figure 1: *The variational family $\mathcal{Q}_A$ for A-VI is a subset of the variational family $\mathcal{Q}_F$ for F-VI. (a) In general, F-VI can achieve a lower KL-divergence than A-VI. (b) Under certain conditions, however A-VI may still achieve the same optimal solution $q^*$ as F-VI.*

expectation-maximization. In this application, $p(\mathbf{x} \mid \theta, \mathbf{z})$ is specified by a deep generative model. The inference function, termed the "encoder", is used to approximate the conditional posterior $p(z_n \mid x_n, \theta)$ for the expectation step. We then estimate $\theta$ by maximizing the approximated marginal likelihood $p(\mathbf{x} \mid \theta)$, which is the maximization step.

There exist several motivations for A-VI. One of them is scaling. While F-VI requires fitting a separate variational factor for each of the data points, A-VI can be more efficient since what we learn about $\phi$ can be amortized across data points. However, if A-VI's inference function is not sufficiently expressive, it may fail to produce as sophisticated a solution as F-VI. We will formalize this intuition and go a little beyond, showing that no matter how expressive the inference function, $\mathcal{Q}_A$ is always a poorer family than $\mathcal{Q}_F$.

While A-VI is typically understood as a cog in the VAE, its formulation suggests a more general algorithm for approximate posterior inference. In this paper, we study A-VI as a general-purpose alternative to F-VI. We ask: Under what conditions can A-VI achieve the same solution as F-VI?

In more detail, because $\mathcal{Q}_A(\mathcal{F})$ is a poorer family than $\mathcal{Q}_F$, A-VI cannot achieve a lower KL-divergence than F-VI's optimal approximation. So, our goal is to distinguish the two scenarios illustrated in Figure 1. In one, the amortized family contains the optimal factorized variational distribution; in the other, the amortized family does not contain it. In the VAE literature, A-VI's potential suboptimality relative to F-VI is known as the *amortization gap* [Cremer et al., 2018].

First, we characterize the class of models where A-VI can close the amortization gap and show that this class corresponds to *simple hierarchical models* [Agrawal and Domke, 2021], i.e. latent variable models which factorize as:

$$p(\theta, \mathbf{z}, \mathbf{x}) = p(\theta) \prod_{n=1}^{N} p(z_n) p(x_n \mid z_n, \theta). \quad (4)$$

This class includes the deep generative model that underlies the VAE and many other models in machine learning and in Bayesian statistics. Our analysis also shows that A-VI is appropriate for *full Bayesian inference*, meaning we approximate $p(\theta, \mathbf{z} \mid \mathbf{x})$, rather than approximate $p(\mathbf{z} \mid \mathbf{x}, \theta)$ and

point-estimate $\theta$ as in variational expectation-maximization.

Second, we generalize A-VI by expanding the domain of the inference function beyond a single data point $x_n$. We establish verifiable conditions for when an expanded function can close the amortization gap, and we provide a time-series example. Finally, we show that there are important examples, such as the hidden Markov model and the Gaussian process, where A-VI cannot attain F-VI's optimal solution, even if expanding the domain of the inference function.

**Plan.** In § 2 we show that the potential for A-VI to achieve F-VI's solution amounts to implicitly solving an *amortization interpolation problem* between $x_n$ and the optimal variational factors of F-VI. For a solution to exist, two conditions must be met: (i) the interpolation problem must be well-posed, which is a condition on the model $p(\theta, \mathbf{z}, \mathbf{x})$; and (ii) the class of inference functions over which we learn $f_\phi$ must be sufficiently expressive, which is a condition on the inference algorithm.

In § 3, we investigate condition (i) theoretically. We show that, in general, the amortization interpolation problem admits a solution if and only if $p(\theta, \mathbf{z}, \mathbf{x})$ is a simple hierarchical model. We then show how to expand the inference function to accommodate more models, and that there are models for which the gap cannot be closed.

In § 4, we empirically study condition (ii). We find that the number of parameters of the inference function does not need to scale with $N$ for A-VI to achieve F-VI's solution, whereas the number of parameters for F-VI must scale with $N$. We demonstrate this phenomenon across several models, including a Bayesian neural network. We also find that when the class of inference functions is sufficiently expressive, A-VI often converges faster than F-VI to the optimal solution. However, in some problems, the performance of A-VI may be much more sensitive to the random seed than F-VI.

**Related work.** The amortization gap has been extensively studied in the context of VAEs [Hjelm et al., 2016, Cremer et al., 2018, Kim et al., 2018, Marino et al., 2018, Krishnan et al., 2018, Kim and Pavlovic, 2021]. This paper goes beyond the VAE, seeking to understand when and how the

amortization gap closes for latent variable models in general. The accuracy of A-VI has also been studied for calculations on held-out likelihoods [Shu et al., 2018]. That said, our focus here is on using A-VI for posterior inference, rather than predictive distributions.

There has been some interest in applying A-VI to models other than standard VAEs [Gershman and Goodman, 2014], including dynamic VAEs [Girin et al., 2021], latent Dirichlet allocation models [Srivastava and Sutton, 2017], and Bayesian hierarchical models [Agrawal and Domke, 2021]. In this paper, we study latent variable models in general, rather than focus on a specific model.

In some applications of A-VI, researchers have expanded the domain of the inference function beyond a single data point. The conventional wisdom is that the inference function should take as input the same data that the exact posterior of the local variable $z_n$ depends on [e.g Girin et al., 2021, chapter 4]. When doing full Bayesian inference, each latent variable $z_n$ typically depends on the entire data set $\mathbf{x}$, however we argue that it can be sufficient to only pass $x_n$ to $f_\phi$. Hence the amortization interpolation problem provides a weaker condition than *a posteriori* dependence on when the amortization gap can be closed.

In addition to passing more data points, it is also possible to pass latent variables to $f_\phi$, notably in hierarchical models [e.g Webb et al., 2018, Agrawal and Domke, 2021, Girin et al., 2021]. This strategy changes the factorization of $q(\theta, \mathbf{z})$ and is aimed at closing the inference gap, i.e. further reducing $\mathrm{KL}(q||p)$ towards 0 or equivalently increasing the evidence lower bound (ELBO), rather than the amortization gap. This type of A-VI is beyond the scope of our paper, though extension of our analysis to such inference functions is feasible.

## 2 PRELIMINARIES

We first set up some theoretical facts about A-VI and F-VI, and articulate the conditions under which the A-VI solution is as accurate as the F-VI solution. We assume that both variational families (Eqs. 2 and 3) use the same type of distribution for $q_0(\theta)$ and so we focus on the variational distributions of $z_n$. For each local latent variable $z_n$, F-VI assigns a marginal distribution $q_n(z_n ; \nu_n)$ from a parametric family $\mathcal{Q}_\ell$ with parameter $\nu_n \in \mathcal{U}$, where $\mathcal{U}$ denotes the space of valid parameters for the variational distribution $\mathcal{Q}_\ell$. The joint family $\mathcal{Q}_F$ is then defined as the product of marginals $q_0(\theta ; \nu_0) \prod_{n=1}^N q_n(z_n ; \nu_n)$. Minimizing the KL-divergence of Eq. 1 yields the optimal variational parameters $\nu^* = (\nu_0^*, \nu_1^*, \cdots, \nu_N^*)$.

Let $\mathcal{X}$ be the space of $x_n$. A-VI fits a function $f_\phi : \mathcal{X} \to \mathcal{U}$ over a family of inference functions $\mathcal{F}$ parameterized by $\phi$ and the KL-divergence of Eq. 1 is minimized with respect to $\phi$. We denote the resulting variational family $\mathcal{Q}_A(\mathcal{F})$.

**Proposition 2.1.** *For any class of inference functions $\mathcal{F}$, $\mathcal{Q}_A(\mathcal{F})$ is a strict subset of $\mathcal{Q}_F$.*

*Proof.* It is straightforward to see from Eq. 2 and Eq. 3 that $\mathcal{Q}_A(\mathcal{F})$ is a subset of $\mathcal{Q}_F$.

To make the ordering strict, it suffices to find an element in $\mathcal{Q}_F$ which does not belong to $\mathcal{Q}_A$. Note this element need not be a minimizer of $\mathrm{KL}(q||p)$. Consider a case where two data points are equal, $x_n = x_m$. Then there exists a distribution $\tilde{q}(\theta, \mathbf{z}) \in \mathcal{Q}_F$ such that $\nu_n \neq \nu_m$, however, we necessarily have $f_\phi(x_n) = f_\phi(x_m)$, and so $\tilde{q}(\theta, \mathbf{z}) \notin \mathcal{Q}_A(\mathcal{F})$. $\square$

An immediate consequence of the ordering is that A-VI cannot achieve a lower KL-divergence than F-VI, leading to a potential amortization gap. To close the gap, the inference function $f_\phi$ must interpolate between $x_n$ and the optimal variational parameter $\nu_n^*$,

$$f_\phi(x_n) = \nu_n^*, \quad \forall n. \tag{5}$$

We call the problem of finding an $f$ that solves Eq. 5 the *amortization interpolation problem.*

**Definition 2.2.** *Given a data set $\mathbf{x}$, suppose $f : \mathcal{X} \to \mathcal{U}$ solves Eq. 5. Then we say $f$ is an ideal inference function.*

The strict ordering between $\mathcal{Q}_A(\mathcal{F})$ and $\mathcal{Q}_F$ warns us that the amortization interpolation problem may not be well posed, since we may find ourselves in a setting where $x_n = x_m$ but $\nu_n^* \neq \nu_m^*$, in which case no ideal inference function exists. In the next section, we derive conditions on the model $p(\theta, \mathbf{z}, \mathbf{x})$ that guarantee the existence of an ideal inference function. Once we establish that the amortization problem is well posed, we can ask how rich does $\mathcal{F}$ need to be to include an ideal inference function. We will investigate this question empirically in § 4.

## 3 EXISTENCE OF AN IDEAL INFERENCE FUNCTION

We first present a lemma that characterizes the optimal variational parameters of F-VI for any model.

**Lemma 3.1.** *(CAVI rule) Consider a probabilistic model $p(\theta, \mathbf{z}, \mathbf{x})$. The optimal solution for F-VI verifies,*

$$q(z_n ; \nu_n^*) \propto \exp\left\{\mathbb{E}_{q(\theta ; \nu_0^*)}\left[\mathbb{E}_{q(\mathbf{z}_{-n} ; \nu^*)}\left[\log p(\theta, \mathbf{z}, \mathbf{x})\right]\right]\right\}, \tag{6}$$

*where $\mathbb{E}_{q(\mathbf{z}_{-n} ; \nu^*)}$ is the expectation with respect to all $z_j$'s except $z_n$.*

*Proof.* See Appendix A.1. $\square$

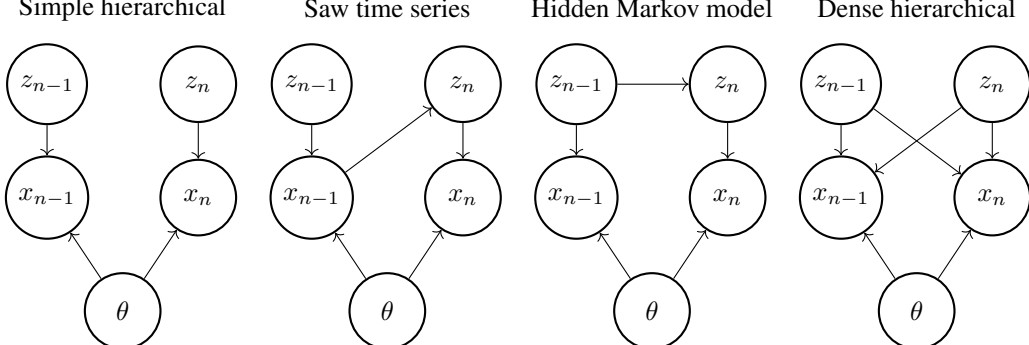

Figure 2: *For the simple hierarchical model (Eq. 4), an ideal inference function $f_\mathbf{x}$ such that $f_\mathbf{x}(x_n) = q(z_n\,;\,\nu_n^*)$ exists. The saw time-series requires learning a map with two inputs $(x_{n-1}, x_n)$. For the Hidden Markov and dense hierarchical graphs, there is no ideal inference function. In the dense hierarchical model, there is an edge between every element of $\mathbf{z}$ and every element of $\mathbf{x}$. For clarity we removed edges between $\theta$ and $z_n$ in all graphs.*

The proof follows from applying the coordinate ascent VI update rule [Blei et al., 2017, Eq. 17] at the optimal solution $\nu^*$. Note that the optimal variational parameters depend on the data and so, where helpful, we write $\nu^* = \nu^*(\mathbf{x})$.

The CAVI rule uses the factorization of $q$ but makes no assumption about the model. We will reason about the factorization of $p(\theta, \mathbf{z}, \mathbf{x})$ using a directed acyclic graph (DAG) representation and define an *exchangeable latent variable model* based on a set of common assumptions.

**Definition 3.2.** *An exchangeable latent variable model $p(\theta, \mathbf{z}, \mathbf{x})$ verifies*

> *(i) local dependence, i.e. there is an edge between $x_n$ and $z_n$ and $p(x_n \mid \mathbf{z}, \theta) \neq p(x_n \mid \mathbf{z}_{-n}, \theta)$.*
>
> *(ii) conditional independence of $x_n$ on $\mathbf{x}_{-n}$ given $\mathbf{z}$ and $\theta$, i.e. $p(\mathbf{x} \mid \mathbf{z}, \theta) = \prod_{n=1}^{N} p(x_n \mid \mathbf{z}, \theta)$.*
>
> *(iii) common distributional forms: no distribution involving $\theta$, $\mathbf{z}$ or $\mathbf{x}$ depends on the index of the random variables.*

Figure 2 presents several graphical models which conform to the above definition, including hierarchical models and certain time series.

**Definition 3.3.** *Following Agrawal and Domke [2021], we define a simple hierarchical model as an exchangeable latent variable model that factorizes according to:*

$$p(\theta, \mathbf{z}, \mathbf{x}) = p(\theta) \prod_{n=1}^{N} p(z_n \mid \theta) p(x_n \mid z_n, \theta).$$

The main result of this section states that the existence of an ideal inference function is, in general, equivalent to $p(\theta, \mathbf{z}, \mathbf{x})$ being a simple hierarchical model.

**Theorem 3.4.** *Consider an exchangeable latent variable model $p(\theta, \mathbf{z}, \mathbf{x})$.*

1. *Suppose $p(\theta, \mathbf{z}, \mathbf{x})$ is a simple hierarchical model. Then an ideal inference function exists.*

2. *Suppose an ideal inference function exists for each $p(\theta, \mathbf{z}, \mathbf{x})$ that factorizes according to a graph. Then this graph is the class of simple hierarchical models.*

*Proof.* See Appendix A.2. $\square$

*Remark* 3.5. The converse in Theorem 3.4 (item 2) is stated for a class of models, meaning the result must hold for any choice of distribution $p(\theta, \mathbf{z}, \mathbf{x})$ supported by the graph. This excludes edge cases that arise due trivial symmetries (see Appendix A.3).

We now provide an outline of the proof for Theorem 3.4. Applying the CAVI rule to the simple hierarchical model, we can show that F-VI's optimal solution takes the form

$$q(z_n\,;\,\nu_n^*) \propto \int_\Theta g(\theta, \mathbf{x})[m(\theta, z_n) + h(\theta, z_n, x_n)]\mathrm{d}\theta, \quad (7)$$

where $g(\theta, \mathbf{x}) = q(\theta\,;\,\nu_0^*(\mathbf{x}))$, $m(\theta, z_n) = \log p(z_n \mid \theta)$ and $h(\theta, x_n) = \log p(x_n \mid z_n, \theta)$. While the function $g$ depends on the entire data set $\mathbf{x}$, it is common to all factors of $q(\mathbf{z})$. Meanwhile, elements specific to $z_n$ only depend on $x_n$. Moreover, $x_n = x_m$ implies $q(z_n\,;\,\nu_n^*) = q(z_m\,;\,\nu_m^*)$. Since a parametric density is uniquely defined by its parameter, we also have a map between $x_n$ and $\nu_n^*$.

We show item (2) by starting with the CAVI rule for a general $p(\theta, \mathbf{z}, \mathbf{x})$ and identifying terms in the kernel of $q(z_n\,;\,\nu_n^*)$ which are not common to all factors of $q(\mathbf{z})$ but depend on elements of $\mathbf{x}$ other than $x_n$. To "eliminate" these

terms, we need to sever edges in the graphical representation of $p(\theta, \mathbf{z}, \mathbf{x})$. Once we remove the offending terms, we are left with a simple hierarchical model.

## 3.1 EXAMPLE: LINEAR PROBABILISTIC MODEL

We now provide an illustrative example where an ideal inference function can be written in closed form. Consider the simple hierarchical model,

$$p(\theta) \propto 1; \ p(z_n) = \mathcal{N}(0, 1); \ p(x_n \mid z_n, \theta) = \mathcal{N}(\theta + \tau z, \sigma^2),$$
(8)

where $\tau \in \mathbb{R}$ and $\sigma \in \mathbb{R}$ are fixed. In this example, the marginal posterior $p(z_n \mid \mathbf{x})$ depends on the entire data set $\mathbf{x}$, rather than on $x_n$ alone, a phenomenon known as *partial pooling* in the Bayesian statistics litterature [Gelman et al., 2013]. This is because rather than hold $\theta$ fixed, we marginalize over it to do full Bayesian inference.

**Proposition 3.6.** *Let $q(z_n; \nu^*)$ be the optimal solution returned by F-VI, when optimizing over the family of factorized Gaussians. Then*

$$\mathbb{E}_{q(z_n; \nu_n^*)}(z_n) = \frac{\tau}{\sigma^2 + \tau^2}(x_n - \bar{x}); \ \mathrm{Var}_{q(z_n; \nu^*)}(z_n) = \xi^2,$$
(9)

*where $\bar{x}$ is the average of $x_{1:N}$, and $\xi$ is a constant.*

*Proof.* See Appendix A.4. □

The bulk of the proof is to work out the posterior $p(\theta, \mathbf{z} \mid \mathbf{x})$ analytically. Note a simple argument of conjugacy does not suffice since we also need to marginalize over $\theta$.

We can rewrite the optimal mean (Eq. 9) as a linear function,

$$\mathbb{E}_{q(z_n; \nu_n^*)}(z_n) = \alpha_0(\mathbf{x}) + \alpha x_n;$$
$$\alpha_0(\mathbf{x}) = -\frac{\tau \bar{x}}{\sigma^2 + \tau^2}; \ \alpha = \frac{\tau}{\sigma^2 + \tau^2}.$$
(10)

For A-VI to match F-VI's solution, we need to learn a linear function for the mean and a constant for the variance, and so, regardless of the number of observations $N$, we can close the amortization gap by learning 3 variational parameters.

This example provides intuition behind Theorem 3.4, which connects A-VI to classical ideas in hierarchical Bayesian modeling. In the considered example, the posterior mean demonstrates partial pooling, a key property of hierarchical models [Gelman et al., 2013]: the posterior mean of $z_n$ depends on both the local observation $x_n$ and on the non-local observations through $\bar{x}$. Even though $p(z_n \mid \mathbf{x}) \neq p(z_n \mid x_n)$, the posterior density of each latent variable is distinguished by the local influence of $x_n$, while the global influence of $\bar{x}$ is the same for all latent variables. As a result $x_n = x_m \implies p(z_n \mid \mathbf{x}) = p(z_m \mid \mathbf{x})$, and an ideal inference function exists.

## 3.2 FURTHER FACTORIZATIONS OF $p(\theta, \mathbf{z}, \mathbf{x})$

Theorem 3.4 tells us that in general, A-VI cannot achieve F-VI's solution for latent variable models other than the simple hierarchical model. We show however that for certain models, it is possible to extend the domain of the inference function in order to close the amortization gap.

A general strategy to verify if the amortization interpolation problem can be solved is to prove the existence of a (potentially expanded) ideal inference function by applying the CAVI rule (Lemma 3.1) to any model $p(\theta, \mathbf{z}, \mathbf{x})$ of interest.

**Saw time series.** Consider the saw time series model,

$$p(\theta, \mathbf{z}, \mathbf{x}) = p(\theta) \prod_{n=1}^{N} p(z_n \mid x_{n-1}) p(x_n \mid z_n, \theta), \quad (11)$$

where each latent variable $z_n$ depends on the previous observation $x_{n-1}$. Applying the CAVI rule, we have

$$q(z_n; \nu_n^*) \propto p(z_n \mid x_{n-1}) \exp\left\{\mathbb{E}_{q(\theta; \nu_0^*)}[\log p(x_n \mid z_n, \theta)]\right\},$$
(12)

which defines a (data-set dependent) function from $(x_{n-1}, x_n)$ to the optimal variational factor. There is no ideal inference function, $f_{\mathbf{x}}: \mathcal{X} \to \mathcal{U}$, however, there exists an ideal inference function $f_{\mathbf{x}}: \mathcal{X} \times \mathcal{X} \to \mathcal{U}$, such that $f_{\mathbf{x}}(x_{n-1}, x_n) = \nu_n^*$ for all $n > 1$.

*Remark* 3.7. When extending the domain of the inference function, we must address edge cases which may not have the requisite argument. For example, inference for $z_1$ requires passing $(x_0, x_1)$ to $f_\phi$, but $x_0$ is not observed. In this case, we assign a distinct variational parameter $\nu_1$ to the factor $q(z_1)$, rather than use amortization.

**Hidden Markov model (HMM).** We now consider another time series, where even after expanding the domain of the inference function, the amortization gap cannot be closed. The joint of the HMM is

$$p(\theta, \mathbf{z}, \mathbf{x}) = p(\theta) \prod_{n=1}^{N} p(z_n \mid z_{n-1}) p(x_n \mid z_n, \theta). \quad (13)$$

The next proposition states that there is in general no ideal inference function $f: \mathcal{X} \to \mathcal{U}$ and furthermore expanding the domain of the inference functions still yields no ideal function.

**Proposition 3.8.** *Consider the HMM of Eq. 13. Let $\mathbf{w}_n \in \mathcal{W}$ be a strict subset of $\mathbf{x}$. There exist HMMs with no ideal inference function $f_{\mathbf{x}}: \mathcal{W} \to \mathcal{U}$. That is, we cannot construct an $f_{\mathbf{x}}$ such that $f_{\mathbf{x}}(\mathbf{w}_n) = \nu_n^*$ for all $n$ which do not constitute an edge case (Remark 3.7).*

*Proof.* See Appendix A.5. □

*Remark* 3.9. If we extend the domain of the inference function to the entire data set $\mathbf{x}$, then A-VI reduces to F-VI and the amortization gap is trivially closed.

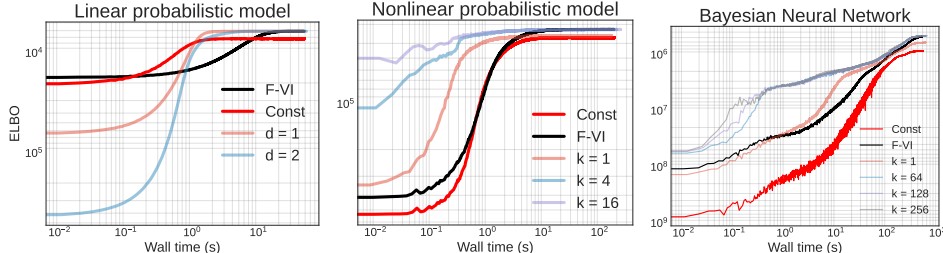

Figure 3: *Examples of optimization paths. As benchmarks, we use F-VI and a constant factor algorithm which assigns the same distribution to all $q(z_n)$. A-VI is then run using different classes of inference functions: (left) we vary the degree $d$ of a learning polynomial; (middle, right) we vary the width $k$ of an inference neural network. For a sufficiently complex inference function, we find that A-VI attains the same ELBO as F-VI, meaning the amortization gap is closed. For results across multiple seeds, see Figure 4.*

The proof of Proposition 3.8 is obtained by constructing a (non-adversarial) example. We argue the above result holds in general and provide a conceptual explanation as to why. In the simple hierarchical and saw time series models the existence of an ideal inference function (respectively over $\mathcal{X}$ and $\mathcal{X} \times \mathcal{X}$) is due to the fact that each data point either has a local or a global influence on $q(z_n\,;\,\nu_n^*)$. In the case of an HMM, there is no common global influence: any observation $x_m$ will have a different influence on the variational factor for each latent variable $z_n$. Moreover, each observation is, to a varying degree, local to any latent variable. A similar reasoning can be applied to the dense hierarchical model (Figure 2), which includes Gaussian process models.

## 4 NUMERICAL EXPERIMENTS

We corroborate our theoretical results on several examples and explore the trade-off between the complexity of the inference function, the quality of the approximation, and the convergence time of the optimization. In all our examples, we do full Bayesian inference over $\theta$ and $\mathbf{z}$. The code to reproduce the experiments can be found at https://github.com/charlesm93/AVI-when-and-why.

### 4.1 EXPERIMENTAL SETUP

As our variational approximation, we use the family of factorized Gaussians. Our benchmarks are a *constant factor algorithm*, which assigns the same Gaussian factor $\bar{q}$ to each latent variable $z_n$, and F-VI. The constant factor and F-VI are respectively the poorest and richest variational families.

With A-VI, we learn $q(\theta)$, just as in F-VI, but amortize the inference for $q(\mathbf{z})$: specifically, we fit an inference function between $x_n$, and the mean and variance of the Gaussian factor $q(z_n)$. For the saw-time series example (§ 4.5) the input of the inference function is expanded to $(x_{n-1}, x_n)$.

To optimize the KL-divergence, we maximize the evidence

lower-bound (ELBO),

$$\mathbb{E}_{q(\mathbf{z}, \theta\,;\,\nu)} \left[ \log p(\theta, \mathbf{z}, \mathbf{x}) - \log q(\theta, \mathbf{z}) \right], \qquad (14)$$

estimated via Monte Carlo. For all experiments we use a conservative 100 draws to estimate the ELBO, except when training a Bayesian neural network, where we use a mini-batching strategy instead.

We employ the Adam optimizer [Kingma and Ba, 2015] in PyTorch [Paszke et al., 2019] and use the reparameterization trick to evaluate the gradients [Kingma and Welling, 2014]. For F-VI, the optimization is directly performed over the parameters of the factorized Gaussian $q(\theta)q(\mathbf{z})$. For A-VI, the optimization is over the parameters of the Gaussian $q(\theta)$ and over the parameters of the inference function (e.g. the weights of the inference neural network) which maps $x_n$ to the parameters of $q(z_n)$. We find a learning rate of 1e-3 works well across applications. The optimizer is stochastic because of the random initialization and the Monte Carlo estimation of the ELBO, and so we repeat each experiment 10 times. Depending on the choice of variational family, the computation cost per optimization step can vary. Therefore we report the ELBO against the wall time when evaluating the performance of each algorithm.

### 4.2 LINEAR PROBABILISTIC MODEL

We begin with the example from § 3.1, using $N = 10,000$ simulated observations, obtained by drawing $\theta$ and $\mathbf{z}$ from standard normals. A-VI's inference function is a polynomial of degree $d$ and we require $d = 1$ to learn the optimal variational parameters (Proposition 3.6). Figure 3 shows the optimization paths over 5,000 steps for a single seed and Figure 4 summarises the number of iterations to converge across seeds for each VI algorithm. Consistent with our analysis in § 3.1, A-VI attains the same outcome as F-VI for $d \geq 1$. Furthermore, we find that A-VI requires an order of magnitude less time to converge. Naturally, using $d = 2$ also yields an optimal solution, however we observe that A-VI then converges more slowly.

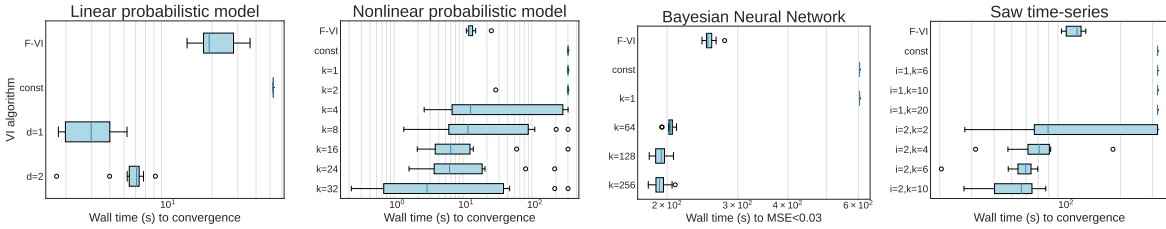

Figure 4: *Wall time to convergence. We run each experiment 10 times and summarize the wall time required for the ELBO to converge for each VI algorithm. For the Bayesian Neural Network, we report convergence in terms of MSE for the image reconstruction. Algorithms with a collapsed box plot on the right do not close the amortization gap.*

## 4.3 NONLINEAR PROBABILISTIC MODEL

This is a variation on the previous model, with a nonlinear likelihood. The joint distribution is then,

$$p(\theta) = \mathcal{N}(0,1)$$
$$p(z_n) = \mathcal{N}(0,1)$$
$$p(x_n \mid z_n, \theta) = \mathcal{N}\left(\theta + z_n(1 + \sin(z_n)), \cos^2(z_n)\right). \tag{15}$$

$N = 10,000$ observations are obtained by simulation. The inference function $f_\phi$ is a neural network with two hidden layers of width $k$ and ReLu activation. A-VI can match F-VI's solution for $k \geq 4$. In contrast to the linear probabilistic example, an overparameterized inference function yields faster convergence as measured by the median over 10 seeds (Figure 4). However, A-VI is more sensitive to the seed and in some cases, can fail to converge after 20,000 iterations. A strength of F-VI relative to A-VI is therefore robustness to initialization, particularly in this example. Given the large number of Monte Carlo samples used to estimate the ELBO, we deduce A-VI is sensitive to the initialization. The choice $k = 16$ produces a fast and reasonably stable algorithm.

## 4.4 BAYESIAN NEURAL NETWORK

Next we consider a deep generative model applied to the FashionMNIST data set [Xiao et al., 2017]. We associate with each image $x_n \in \mathbb{R}^{784}$ a low-dimensional representation $z_n \in \mathbb{R}^{64}$. The joint distribution is then,

$$p(\theta) = \mathcal{N}(0, I)$$
$$p(z_n) = \mathcal{N}(0, I)$$
$$p(x_n \mid z_n, \theta) = \mathcal{N}(\Omega(z_n\,;\,\theta), I), \tag{16}$$

where $\Omega$ is a neural network with two hidden layers of width 256 and a leaky ReLu activation function. $\theta \in \mathbb{R}^{57,232}$ stores the weights and biases of the network. This generative model underlies the traditional VAE, however we estimate a posterior over $\theta$ in order to confirm that A-VI can indeed close the amortization gap when fitting a Bayesian neural network. We train the model on 10,000 images and at each

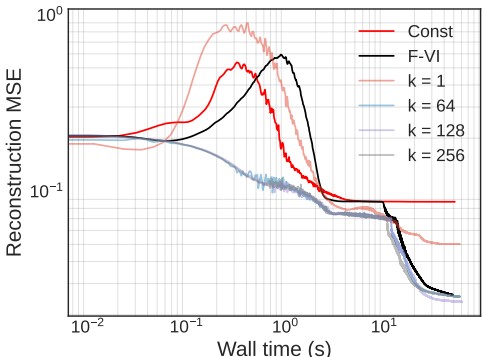

Figure 5: *Image reconstruction error, as measured by MSE over pixel, for a trained Bayesian neural network. The MSE is not a one-to-one map with the ELBO. For a sufficiently expressive inference network, A-VI achieves the same error as F-VI and converges faster. The above provides the paths for a single seed; for results across several seeds, see Figure 4.*

iteration, evaluate the ELBO on a mini-batch of 1,000 images. Hence a single epoch contains 10 iterations, and we run each VI algorithm for 5,000 epochs. From a pilot run, we found estimating the ELBO with a single Monte Carlo sample worked reasonably well.

Due to the non-linear landscape of the optimization, we cannot guarantee that any of the algorithms converge. However, we find that after 5,000 epochs, A-VI achieves the same ELBO as F-VI when using a width $k \geq 64$ for the inference network (Figure 3). We also study the image reconstruction error measured by the mean squared error (MSE) over pixels on the training set. (The MSE on the test set is not available for F-VI, however in Appendix B we provide test error for A-VI.) For this calculation, we use the Bayes estimator $\mathbb{E}(\theta \mid \mathbf{x})$ and $\mathbb{E}(\mathbf{z} \mid \mathbf{x})$. Figure 5 plots the MSE against wall time for the same seed used in Figure 3. In Figure 4, we report the wall time required to achieve an MSE below 0.03, which corresponds to F-VI's best solution. For $k \geq 64$, A-VI requires 2-3 times less iterations to converge, however each iteration is considerably more expensive. As a result, the speed-up when examining wall-time is ∼25%. Overpa-

rameterizing the inference network slightly improves the convergence speed.

## 4.5 SAW TIME SERIES

In this final example, we explore the benefits of extending the domain of the inference function. We simulate $N = 1,000$ observations from a saw time series (eq. 11), with $x_0 = 0$ and

$$p(\theta) = \mathcal{N}(0, 1)$$
$$p(z_0) = \mathcal{N}(0, 1)$$
$$p(z_n \mid x_{n-1}) = \mathcal{N}(x_{n-1}, 1);$$
$$p(x_n \mid z_n) = \mathcal{N}(\alpha(\theta + z_n), 1). \quad (17)$$

Once again, we fit an inference neural network with two hidden layers of width $k$. Additionally, we allow the network to either take in $x_n$ or $(x_{n-1}, x_n)$ as its input. Only with the expanded output does A-VI attain F-VI's optimum for $k \geq 4$. Using only $x_n$ produces a suboptimal approximation even with a comparatively large inference network (e.g. $k = 20$). Figure 6 demonstrates this behavior for one optimization path. Across seed, we find that A-VI consistently outperforms F-VI (Figure 4). While A-VI is sensitive to the seed for $k = 2$, the algorithm stabilizes once we overparameterize the inference network, with $k \geq 4$.

## 5 DISCUSSION

We studied amortized variational inference (A-VI) as a general method for posterior approximation. We derived a necessary, sufficient, and verifiable condition on the model $p(\theta, \mathbf{z}, \mathbf{x})$ under which A-VI can achieve the same optimal solution as factorized (or mean-field) variational inference (F-VI). These results establish that A-VI is a viable method for a large class of hierarchical models, including when doing a full Bayesian analysis rather than using a point estimator for the global parameter $\theta$.

We then examined how to extend the domain of the inference function for models beyond the simple hierarchical model. We also established that there are some models for which the amortization gap cannot be closed, even after expanding the domain of the inference function and no matter how expressive the inference function is. In such models, our results can provide justification for methods such as semi-amortized VI [Kim et al., 2018, Kim and Pavlovic, 2021], in which A-VI is used to converge quickly to a suboptimal solution, which is then refined with F-VI.

There remain several open questions about amortized variational inference.

Even when the model admits an ideal inference function, a persistent question is how to choose the class of inference functions in order to close the amortization gap. The

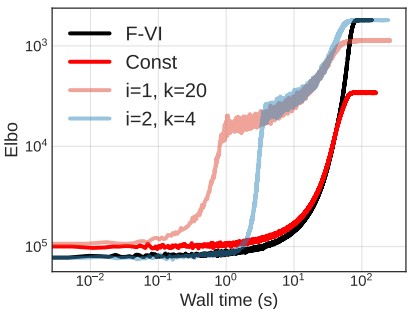

Figure 6: *Optimization path for saw time series. An inference network which only takes in $x_n$ as its input ($i = 1$ case) cannot close the amortization gap, even when using a relatively large network. On other hand, a network which takes in $x_{n-1}, x_n$ ($i = 2$ case) closes the gap with a relatively small network.*

ordering of the variational families $\mathcal{Q}_A \subset \mathcal{Q}_F$ suggests an informal diagnostic: after A-VI converges, run a few steps of F-VI and see if the solution improves. If it does then the inference function may not be sufficiently expressive to close the gap. It is also possible that the optimizer converged to a local optimum and that changing the variational objective allows the solution to improve. On the other hand, for high-dimensional problems with highly non-convex landscapes, it may take many iterations before the solution improves, in which case the proposed diagnostic would not detect shortcomings in the inference function.

A related question: For a choice of the class of inference functions, how does A-VI change the optimization landscape relative to F-VI? Our experimental results also raise the question of whether an overparameterized class of inference functions burdens the optimization, as seen for the linear probabilistic model, or improves the convergence rate, as illustrated in the Bayesian neural network and the saw time series. Along similar lines, it is of interest to study the advantages and drawbacks of A-VI on more complex data sets than the ones we have considered, particular case for which convergence may not be achieved within a reasonable computational budget.

Finally, how accurate is A-VI when applied to held-out data? We expect the *generalization gap* [Shu et al., 2018, Ganguly et al., 2022] can also be analyzed by setting up an implicit interpolation problem, this time with constraints to not overfit the data. In a full Bayesian context, how can we understand the role of A-VI for online learning? In other words, how well can $f_\phi(x_{N+1})$ approximate $p(z_n \mid \mathbf{x}, x_{N+1})$?

## 6 ACKNOWLEDGMENT

We thank Lawrence Saul, Robert Gower, Justin Domke, and Ruben Ohana for helpful discussions, and Achille Nazaret

for feedback on an early version manuscript.

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

# A MISSING PROOFS

We provide proofs for all statements in §3.

## A.1 CAVI RULE

Lemma 3.1 follows from the coordinate-ascent VI update rule for F-VI [Blei et al., 2017, Eq. 17], which tells us how to choose $q(z_n \, ; \, \nu_n)$ to minimize the KL-divergence, while maintaining the other factors in the approximating distribution fixed. Specifically, suppose $\nu_0$ and $\boldsymbol{\nu}_{-n}$ are fixed. Then the optimal variational parameter $\nu_n^\star$ for $n^{\text{th}}$ factor verifies

$$q(z_n \, ; \, \nu_n^\star) \propto \exp \left\{ \mathbb{E}_{q(\theta \, ; \, \nu_0)} \left[ \mathbb{E}_{q(\mathbf{z}_{-n} \, ; \, \nu)} \left[ \log p(\theta, \mathbf{z}, \mathbf{x}) \right] \right] \right\}. \tag{18}$$

We now apply this rule to the optimal solution, i.e. we set $\nu_0 = \nu_0^*$ and $\boldsymbol{\nu}_{-n} = \boldsymbol{\nu}_{-n}^*$. Then, minimizing the KL-divergence, $\nu_n^\star = \nu_n^*$ and the desired result follows. $\qquad\square$

## A.2 EXISTENCE OF AN IDEAL INFERENCE FUNCTION AND SIMPLE HIERARCHICAL MODELS

Theorem 3.4 states that the existence of an ideal inference function for a standard latent variable model (Definition 3.2) is, in general, equivalent to $p(\theta, \mathbf{z}, \mathbf{x})$ being a simple hierarchical model (Eq. 1).

We first prove item (1). Suppose $p(\theta, \mathbf{z}, \mathbf{x})$ is a simple hierarchical model. Applying the CAVI rule (Lemma 3.1) to Eq. 1,

$$q(z_n \, ; \, \nu^*) \propto \exp \left\{ \mathbb{E}_{q(\theta \, ; \, \nu_0^*)} \left[ \mathbb{E}_{q(\mathbf{z}_{-n} \, ; \, \nu^*)} \left[ \log p(\theta) + \sum_{j=1}^n \log p(z_j \mid \theta) + \log p(x_j \mid z_j, \theta) \right] \right] \right\}$$

$$\propto \exp \left\{ \mathbb{E}_{q(\theta \, ; \, \nu_0^*)} \left[ \mathbb{E}_{q(\mathbf{z}_{-n} \, ; \, \nu^*)} \left[ \log p(z_n \mid \theta) + \log p(x_n \mid z_n, \theta) \right] \right] \right\}$$

$$\propto \exp \left\{ \mathbb{E}_{q(\theta \, ; \, \nu_0^*)} \left[ \log p(z_n \mid \theta) + \log p(x_n \mid z_n, \theta) \right] \right\}.$$

Then

$$q(z_n \, ; \, \nu^*) = k_{\mathbf{x}}(x_n) \int_\Theta q(\theta \, ; \, \nu_0^*(\mathbf{x})) \log p(z_n \mid \theta) + \log p(x_n \mid z_n, \theta) \mathrm{d}\theta, \tag{19}$$

where $k_{\mathbf{x}}(x_n) = \left[ \int_{\mathcal{Z}} \int_\Theta q(\theta \, ; \, \nu_0^*(\mathbf{x})) \log p(z_n \mid \theta) + \log p(x_n \mid z_n, \theta) \mathrm{d}\theta \mathrm{d}z_n \right]^{-1}$ is a normalizing constant. The R.H.S of Eq. 19 defines an ideal inference function $f_{\mathbf{x}}(x_n)$, in the sense that, given $\mathbf{x}$, we have $x_n = x_m \implies f_{\mathbf{x}}(x_n) = f_{\mathbf{x}}(x_m)$.

Next we prove the converse, which is item (2) of Theorem 3.3. Applying the CAVI rule to a standard latent variable model,

$$q(z_n \, ; \, \nu^*) \propto \exp \left\{ \mathbb{E}_{q(\theta, \mathbf{z}_{-n} \, ; \, \boldsymbol{\nu}_{-n}^*)} \log p(\theta, \mathbf{z}, \mathbf{x}) \right\}$$

$$\propto \exp \left\{ \mathbb{E}_{q(\theta, \mathbf{z}_{-n} \, ; \, \boldsymbol{\nu}_{-n}^*)} \log p(z_n \mid \mathbf{z}_{-n}, \theta) + \log p(x_n \mid z_n, \mathbf{z}_{-n}, \theta) + \sum_{i \neq n} \log p(x_i \mid z_n, \mathbf{z}_{-n}, \theta) \right\}. \tag{20}$$

The last equation highlights all the terms in which $z_n$ appears. Furthermore, we used the property of *conditional independence* (Definition 3.2 (ii)) to break up the log likelihood $\log p(\mathbf{x} \mid \mathbf{z}, \theta)$ into a sum.

Suppose now that there exists a graph $\mathcal{G}$, such that for *any* standard latent variable model supported by this graph, there exists an ideal inference function, that is $\nu_n^* = f_{\mathbf{x}}(x_n)$. Because $q$ is parametric, we have that the R.H.S of Eq. 20 is also a (dataset dependent) function of $x_n$. For this assumption to hold *for any choice of distribution*, any contribution of $x_{i \neq n}$ that is not common to all the variational factors of $q(\mathbf{z})$ must be absorbed into the normalizing constant and effectively vanish. We will complete the proof by removing unique contributions of $x_i$ and severing offending edges in $\mathcal{G}$ (Figure 7).

The most obvious contribution of $x_i$ appears in the likelihood terms and is removed if and only if we exclude non-local dependence, that is for $i \neq n$, $p(x_i \mid z_n, \mathbf{z}_{-n}, \theta) = p(x_i \mid \mathbf{z}_{-n}, \theta)$. Doing so for every $n$, we have

$$p(x_i \mid z_n, \mathbf{z}_{-n}, \theta) = p(x_i \mid z_i, \theta). \tag{21}$$

*Remark* A.1. Here the assumption of *local dependence* (Definition 3.2 (i)) is critical. Without it, we cannot exclude the possibility that $x_i$ does not depend on $z_i$, or any $z_j$'s other than $z_n$, and hence that $p(x_i \mid z_n, \mathbf{z}_{-n}, \theta) = p(x_i \mid z_n, \theta)$, $i \neq n$. Then an edge between $z_n$ and $x_i$ would not contradict the existence of an ideal inference function.

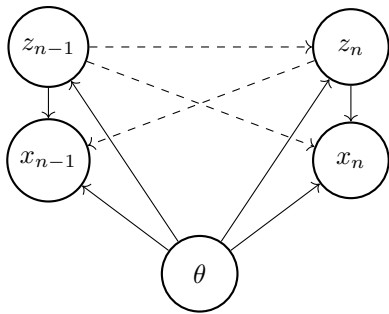

Figure 7: *Graphical representation of a standard latent variable model. If present, the dotted edges preclude the existence of an ideal inference function $f_{\mathbf{x}}(x_n) = \nu_n^*$ and the amortization gap cannot be closed.*

Next, we have by assumption that $\nu_i^* = f_{\mathbf{x}}(x_i)$. Then

$$q(z_n \, ; \, \nu^*) \propto \exp\left\{ \int_{\Theta, \mathcal{Z}_{-n}} q(\mathrm{d}\theta \, ; \, \nu_0(\mathbf{x})) \prod_{i \neq n} q(\mathrm{d}z_i \, ; \, f_{\mathbf{x}}(x_i)) \log p(z_n \mid \mathbf{z}_{-n}, \theta) + \log p(x_n \mid z_n, \theta) \right\}. \tag{22}$$

The offending terms are now the variational factors $q(\mathrm{d}z_i \, ; \, f_{\mathbf{x}}(x_i))$ in the integral. To remove them, we must get rid of any term that couples $z_n$ and $z_i$, and so $z_n$ must be a priori independent of $z_i$, that is

$$p(z_n \mid \mathbf{z}_{-n}, \theta) = p(z_n \mid \theta). \tag{23}$$

A standard latent variable model that verifies Eq. 21 and Eq. 23 must also verify Eq. 1 and is therefore a simple hierarchical model. $\qquad\square$

## A.3  EXAMPLE OF A LATENT VARIABLE MODEL, WHICH IS NOT A SIMPLE HIERARCHICAL MODEL AND ADMITS AN IDEAL INFERENCE FUNCTION

The statement of Theorem 3.4, item (ii) is carefully written for all distributions supported on a graph. To see why a simple "if and only if" version of item (i) is not true, consider a dense hierarchical model, with edges between all elements of $\mathbf{x}$ and $\mathbf{z}$. If we a choose a likelihood which is symmetric in $\mathbf{z}$, e.g. $p(x_n \mid \mathbf{z}, \theta) = p(x_n \mid \sum_n z_n, \theta)$, then there exists a (constant) ideal inference function and moreover, all factors $q(z_n \, ; \, \nu_n^*)$ are identical.

This case is of course trivial: with such a symmetry, the notion of a local latent variable is unjustified. To our knowledge, all examples of models, which are not simple hierarchical models and still admit an ideal inference function, rely on a similar trivialities. These however constitute edge cases we must be mindful of when writing formal statements.

## A.4  ANALYTICAL RESULTS FOR THE LINEAR PROBABILISTIC MODEL

We prove Proposition 3.6, which provides an exact expression for the mean and variance of $q(z_n \, ; \, \nu^*)$, the optimal solution returned by F-VI when applied to the linear generative model. In the model of interest, $\theta$ is a scalar random variable, and we introduce the fixed standard deviations, $\tau \in \mathbb{R}$ and $\sigma \in \mathbb{R}$. Next

$$p(\theta) \propto 1; \; p(z_n) = \mathcal{N}(0, 1); \; p(x_n) = \mathcal{N}(\theta + \tau z_n, \sigma). \tag{24}$$

Since the posterior distribution $p(\theta, \mathbf{z} \mid \mathbf{x})$ is normal, $q(z_n \, ; \, \nu^*)$ can be worked out analytically [e.g Turner and Sahani, 2011, Margossian and Saul, 2023]. Specifically,

$$q(z_n \, ; \, \nu_n^*) = \mathcal{N}\left( \mu_n, \frac{1}{[\Sigma^{-1}]_{nn}} \right), \tag{25}$$

where $\mu_n$ is the correct posterior mean for $z_n$ and $\Sigma$ is the correct posterior covariance matrix. Note that F-VI always underestimates the posterior marginal variance unless $\Sigma$ is diagonal [Margossian and Saul, 2023, Theorem 3.1]. It remains to find an analytical expression for the posterior distribution.

**Lemma A.2.** *The marginal posterior distribution is given by*

$$p(z_n \mid \mathbf{x}) = \mathcal{N}\left(\frac{\tau}{\sigma^2 + \tau^2}(x_n - \bar{x}), s\right), \tag{26}$$

*for some $s$, constant with respect to $\mathbf{x}$.*

*Proof.* From Bayes' rule

$$
\begin{aligned}
\log p(\mathbf{z}, \theta \mid \mathbf{x}) &= k - \frac{1}{2}\sum_{n=1}^{N} z_n^2 - \frac{1}{2\sigma^2}\sum_{n=1}^{N}(x_n - \theta - \tau z_n)^2 \\
&= k - \frac{1}{2}\sum_{n=1}^{N} z_n^2 - \frac{1}{2\sigma^2}\sum_{n=1}^{N}\theta^2 + (x - \tau z_n)^2 - 2\theta(x_n - \tau z_n) \\
&= k - \frac{1}{2}\sum_{n=1}^{N} z_n^2 - \frac{1}{2\sigma^2}\left(n\theta^2 + \sum_{n=1}^{N}(x_n - \tau z_n)^2 - 2\theta\sum_{n=1}^{N}(x_n - \tau z_n)\right),
\end{aligned}
\tag{27}
$$

where $k$ is a constant with respect to $\mathbf{z}$ and $\theta$. Moving forward, we overload the notation for $k$ to designate any such constant. As expected, Eq. 27 is quadratic in $\theta$ and $\mathbf{z}$.

*Remark* A.3. At this point, the proof may take two directions: in one, we work out the precision matrix, $\Phi$ (i.e. the inverse covariance matrix $\Sigma$) for $p(\mathbf{z}, \theta \mid \mathbf{x})$ and invert it to obtain the posterior mean for each $z_n$. Constructing $\Phi$ is straightforward and necessary to show the covariance of $q(z_n \, ; \, \nu_n^*)$ is constant with respect to $\mathbf{x}$. However, inverting $\Phi$ requires recursively applying the Sherman-Morrison formula three times, which is algebraically tedious. The other direction is to marginalize out $\theta$. We can then construct the precision matrix $\Psi$ for $p(\mathbf{z} \mid \mathbf{x})$, which only requires a single application of the Sherman-Morrison formula to invert. We opt for the second direction, noting both options are rather involved.

To marginalize out $\theta$, we complete the square and perform a Gaussian integral,

$$
\begin{aligned}
\log p(\mathbf{z}, \theta \mid \mathbf{x}) &= k - \frac{1}{2}\sum_{n=1}^{N} z_n^2 - \frac{n}{2\sigma^2}\left[\theta^2 + \frac{1}{n}\sum_{n=1}^{N}(x_n - \tau z_n)^2 - 2\theta\sum_{n=1}^{N}(x_n - \tau z_n)\right. \\
&\quad \left. + \left(\frac{1}{n}\sum_{n=1}^{N}(x_n - \tau z_n)\right)^2 - \left(\frac{1}{n}\sum_{n=1}^{N}(x_n - \tau z_n)\right)^2\right] \\
&= k - \frac{1}{2}\sum_{n=1}^{N} z_n^2 - \frac{n}{2\sigma^2}\left[\left(\theta - \frac{1}{n}\sum_{n=1}^{N}(x_n - \tau z_n)\right)^2 + \frac{1}{n}\sum_{n=1}^{N}(x_n - \tau z_n)^2\right. \\
&\quad \left. - \left(\frac{1}{n}\sum_{n=1}^{N}(x_n - \tau z_n)\right)^2\right]
\end{aligned}
\tag{28}
$$

Then

$$\log p(\mathbf{z} \mid \mathbf{x}) = k - \frac{1}{2}\sum_{n=1}^{N} z_n^2 - \frac{1}{2\sigma^2}\left[\sum_{n=1}^{N}(x_n - \tau z_n)^2 - \frac{1}{n}\left(\sum_{n=1}^{N}(x_n - \tau z_n)\right)^2\right]. \tag{29}$$

Expanding the square,

$$\left(\sum_{n=1}^{N}(x_n - \tau z_n)\right)^2 = \sum_{n=1}^{N}(x_n - \tau z_n)^2 + 2\sum_{j<n}(x_n - \tau z_n)(x_j - \tau z_j). \tag{30}$$

Plugging this in and factoring out $\tau$, we get

$$\log p(\mathbf{z} \mid \mathbf{x}) = k - \frac{1}{2}\sum_{n=1}^{N} z_n^2 - \frac{\tau^2}{2\sigma^2}\left[\sum_{n=1}^{N}\left(1 - \frac{1}{n}\right)\left(\frac{x_n}{\tau} - z_n\right)^2 - \frac{2}{n}\sum_{j<n}\left(\frac{x_n}{\tau} - z_n\right)\left(\frac{x_j}{\tau} - z_j\right)\right]. \tag{31}$$

Now the standard expression for a multivariate Gaussian is

$$\log p(\mathbf{z} \mid \mathbf{x}) = k - \frac{1}{2}(\mathbf{z} - \boldsymbol{\mu})^T \Psi (\mathbf{z} - \boldsymbol{\mu}) = k - \frac{1}{2}\left( \sum_{n=1}^{N} \Psi_{nn}(z_n - \mu_n)^2 + 2\sum_{j<n} \Psi_{jn}(z_n - \mu_n)(z_j - \mu_j) \right), \quad (32)$$

where $\boldsymbol{\mu}$ is the mean and $\Psi$ the precision matrix. We solve for the mean and precision matrix by matching the coefficients in the above two expressions for $z_n$, $z_n z_j$, and $z_n^2$, which respectively produce the following equations:

$$\sum_{j=1}^{N} \Psi_{nj}\mu_j = \frac{\tau}{\sigma^2}(x_n - \bar{x}) \qquad (33)$$

$$\Psi_{nj} = -\frac{\tau^2}{n\sigma^2}, \quad \forall n \neq j \qquad (34)$$

$$\Psi_{nn} = 1 + \frac{\tau^2}{\sigma^2}\left(1 - \frac{1}{N}\right). \qquad (35)$$

This immediately gives us the precision matrix. Eq. 33 may be rewritten in matrix form as

$$\boldsymbol{\mu} = \frac{\tau}{\sigma^2}\Psi^{-1}[\mathbf{x} - \bar{x}\mathbf{1}], \qquad (36)$$

where $\mathbf{1}$ is the $N$-vector of 1's. Let $\alpha = \Psi_{nj}$, for any $n \neq j$, and $\beta = \Psi_{nn} - \alpha$. Then

$$\Psi = \beta I + \alpha \mathbf{1}\mathbf{1}^T, \qquad (37)$$

Applying the Sherman-Morrison formula, we obtain the covariance matrix,

$$\begin{aligned} \Psi^{-1} &= (\beta I + \alpha \mathbf{1}\mathbf{1}^T)^{-1} \\ &= \beta^{-1}I - \frac{\beta^{-1}I\alpha \mathbf{1}\mathbf{1}^T \beta^{-1}I}{1 + \alpha \mathbf{1}^T \beta^{-1}I\mathbf{1}} \\ &= \beta^{-1}I - \frac{\alpha\beta^{-1}}{\beta + N\alpha}\mathbf{1}\mathbf{1}^T. \end{aligned} \qquad (38)$$

Notice that $\Psi^{-1}$ does not depend on $\mathbf{x}$ and that it's diagonal elements are all equal. Moreover $(\Psi^{-1})_{nn}$ gives us the constant, $s$. Next let

$$a = \beta^{-1}\frac{\tau}{\sigma^2}; \quad b = -\frac{\alpha\beta^{-1}}{\beta + N\alpha}\frac{\tau}{\sigma^2}. \qquad (39)$$

Then $\boldsymbol{\mu} = (aI + b\mathbf{1}\mathbf{1}^T)[\mathbf{x} - \bar{x}\mathbf{1}\mathbf{1}^T]$ and moreover

$$\begin{aligned} \mu_n &= a(x_n - \bar{x}) + b\sum_{j=1}^{N} x_j - \bar{x} \\ &= a(x_n - \bar{x}) \\ &= \frac{\tau}{\sigma^2}\left(\frac{\tau^2 + \sigma^2}{\sigma^2}\right)^{-1}(x_n - \bar{x}) \\ &= \frac{\tau}{\sigma^2 + \tau^2}(x_n - \bar{x}), \end{aligned}$$

as desired.

$\square$

To complete the proof of Proposition 3.4, we need to show that the variances of $q(z_n ; \nu^*)$ is constant with respect to $\mathbf{x}$; that they are equal for each $z_n$ follows from the symmetry of the problem. We already constructed the precision matrix $\Psi$ for $p(\mathbf{z} \mid \mathbf{x})$, but we actually need to study the full precision matrix $\Phi$ of $p(\theta, \mathbf{z} \mid \mathbf{x})$. We use the index 0 to denote the columns (or rows) corresponding to $\theta$.

**Lemma A.4.** *The posterior precision matrix $\Phi$ of $p(\theta, \mathbf{z} \mid \mathbf{x})$ verfies*

$$\Phi_{00} = \frac{N}{\sigma^2}; \quad \Phi_{0j} = \frac{\tau}{2\sigma^2} \text{ if } j > 0; \quad \Phi_{nn} = 1 + \frac{\tau^2}{\sigma^2} \text{ if } i > 0; \quad \Phi_{nj} = 0, \text{ if } n \neq j. \tag{40}$$

*Crucially, $\Phi$ is constant with respect to $\mathbf{x}$.*

*Proof.* Consider Eq. 27, rewritten here for convenience,

$$\log p(\mathbf{z}, \theta \mid \mathbf{x}) = k - \frac{1}{2} \sum_{n=1}^{N} z_n^2 - \frac{1}{2\sigma^2} \left( N\theta^2 + \sum_{n=1}^{N} (x_n - \tau z_n)^2 - 2\theta \sum_{n=1}^{N} (x_n - \tau z_n) \right).$$

The standard Gaussian form is

$$
\begin{aligned}
\log p(\mathbf{z}, \theta \mid \mathbf{x}) &= k - \frac{1}{2} \Bigg[ \Phi_{00}(\theta - \nu)^2 + \sum_{n=1}^{N} \Phi_{nn}(z_n - \mu_n)^2 \\
&\quad + 2 \left( \sum_{j=1}^{N} \Phi_{0j}(\theta - \nu)(z_j - \mu_j) + \sum_{j<n} \Phi_{nj}(z_n - \mu_n)(z_j - \mu_j) \right) \Bigg].
\end{aligned}
\tag{41}
$$

Matching coefficients for $\theta^2$, $\theta z_j$, $z_n z_j$ and $z_n^2$, we obtain respectively

$$\Phi_{00} = \frac{N}{\sigma^2}; \quad \Phi_{0j} = \frac{\tau}{2\sigma^2} \text{ if } j > 0; \quad \Phi_{nn} = 1 + \frac{\tau^2}{\sigma^2} \text{ if } n > 0; \quad \Phi_{nj} = 0, \text{ if } n \neq j.$$

$\square$

The variance of $q(z_n\,;\,\nu^*)$ is obtained by inverting the diagonal elements of $\Phi$. By symmetry, $\mathrm{Var}_{q^*}(z_n) = \xi \;\; \forall n$, where $\xi$ is a constant which does not depend on $\mathbf{x}$. This completes the proof of Proposition 3.4. $\square$

## A.5 NON-EXISTENCE OF AN IDEAL INFERENCE FUNCTION FOR HIDDEN MARKOV MODELS

To prove Proposition 3.8, we construct an example for which the optimal F-VI solution, using a factorized Gaussian approximation, can be written in a nearly closed form, and show that the optimal variational factors $\nu_n^*$ take different values even when all the values of $\mathbf{x}$ are equal. Then for any strict subset $\mathbf{w}_n \in \mathbf{x}$, we have $\mathbf{w}_n = \mathbf{w}_m$ but $\nu_n^* \neq \nu_m^*$. This provides our counter-example.

Consider the model

$$p(z_0) \propto 1\,;\, p(z_n \mid z_{n-1}) = \mathcal{N}(z_{n-1}, 1)\,;\, p(x_n \mid z_n) = \mathcal{N}(z_n, 1), \tag{42}$$

where $\theta$ is held fixed, say to a point estimate $\hat{\theta}$, and ignored for the rest of this analysis. Applying Bayes' rule and expanding

$$
\begin{aligned}
\log p(\mathbf{z} \mid \mathbf{x}) &= k - \frac{1}{2} \sum_{n=1}^{N} (z_n - z_{n-1})^2 + (x_n - z_n)^2 \\
&= -\frac{1}{2} \sum_{n=1}^{N} 2z_n^2 + z_{n-1}^2 - 2x_n z_n - 2z_n z_{n-1},
\end{aligned}
$$

which is a quadratic form in $\mathbf{z}$ and hence a Gaussian. Matching the coefficients for $z_n$, $z_n z_j$ and $z_n^2$ to the standard expression for a multivariate Gaussian (Eq. 41), we get

$$\sum_{j=1}^{N} \Psi_{nj}\mu_j = -2x_n \tag{43}$$

$$\Psi_{nj} = -2 \quad \text{if } j = n - 1 \text{ or } j = n + 1 \tag{44}$$

$$\Psi_{nn} = 3 \quad \text{if } n \geq 1 \tag{45}$$

$$\Psi_{00} = 1. \tag{46}$$

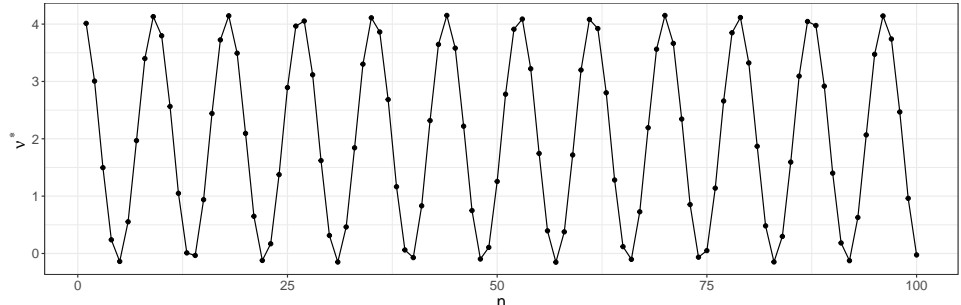

Figure 8: *Optimal variational means when using a Gaussian F-VI on a hidden Markov model (Eq. 42). Even though the elements of $\mathbf{x}$ are all equal, the optimal variational means take on different values and so no inference function $f_\phi : \mathbf{w_n} \to \nu_\mathbf{n}^*$ can be constructed, for any subset $\mathbf{w_n} \in \mathbf{x}$.*

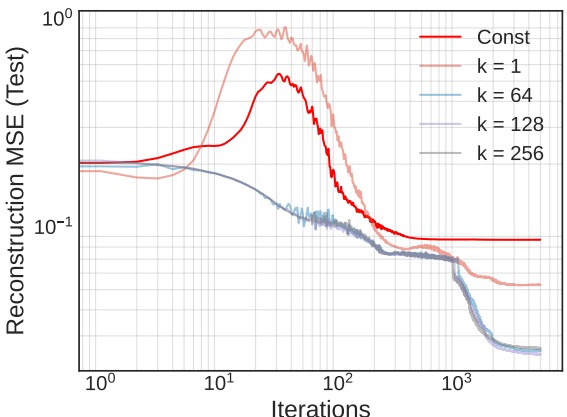

Figure 9: *Reconstruction MSE on a test set.*

All non-specified elements of $\Psi$ go to 0. Moreover the precision matrix $\Psi$ is tri-diagonal. The posterior mean solves the linear problem,

$$\boldsymbol{\mu} = -2\Psi^{-1}\mathbf{x}. \tag{47}$$

Since the variational family and the target are both Gaussian, the optimal variational mean is simply the posterior mean and $\nu^* = \boldsymbol{\mu}$. Even though the elements of $\mathbf{x}$ are all equal, it is in general not the case that the elements of $\nu^*$ are constant. To see this explicitly, we take $N = 100$ and $x_1 = x_2 = \cdots = x_N = 1$, and find that the elements of $\nu^*$ are indeed distinct (Figure 8). This shows that there exists a hidden Markov model and a realization of the data $\mathbf{x}$ such that no learnable inference function exists. $\qquad\square$

## B  ADDITIONAL EXPERIMENTAL RESULTS

**Hardware.** All experiments are conducted in `Python` 3.9.15 with `PyTorch` 1.13.1 and `CUDA` 12.0 using an NVIDIA RTX A6000 GPU.

**Reconstruction error on test set for Bayesian neural network.** We consider the reconstruction error on a test set of 10,000 images (Figure 9). The reconstructed image is obtained by (i) computing $q(z' \mid x')$ using the inference function $f_\phi$ and (ii) feeding $\mathbb{E}_q(z' \mid x')$ into the likelihood neural network $\Omega$ (in the VAE context, the "decoder") to obtain $\hat{x}'$. $\Omega$ is evaluated at the Bayes estimator $\hat{\theta} = \mathbb{E}_q(\theta \mid \mathbf{x})$. F-VI provides no automatic way of doing step (i) (one would need to learn $q(z' ; \nu')$ by running F-VI from scratch), and so we do not evaluate it on the test set. Overall, we find the model generalizes well, and the test error is very close to the training error.

