# OpenReview forum: "Amortized Variational Inference: When and Why?"
_auai.org/UAI/2024/Conference — UAI 2024 poster_

### Official Review · Reviewer_e8MW · 2024-03-20

**Q2-1 Originality-Novelty:** 3
**Q2-2 Correctness-Technical Quality:** 4
**Q2-5 Clarity Of Writing:** 4

**Q1 Summary And Contributions:**

This paper investigate when and why amortized variational inference (A-VI) can attain the optimal solution for factorized variational inference (F-VI) and thereby closes the amortization gap. It consider different latent variable models and also provides an example of such when the amortization gap cannot be closed, the theoretical results are demonstrated by simple experiments.

**Q2-3 Extent To Which Claims Are Supported By Evidence:**

2: Fair: the main claims are somewhat supported by evidence (but the experimental evaluation may be weak, or does not match entirely with the claims, important baselines may be missing, proofs contain important ideas but lack rigor, algorithmic details are only discussed superficially, references are imprecise, assumptions are not sufficiently motivated or explicated, etc.).

**Q2-4 Reproducibility:**

4: Excellent: key resources (e.g. proofs, code, data) are available and key details (e.g. proof sketches, experimental setup) are comprehensively described for competent researchers to confidently and easily reproduce the main results.

**Q3 Main Strengths:**

- theoretical insights on the amortization gap of A-VI w.r.t. F-VI
- clarity of presentation
- rigor of proofs, not restricted to simple argumentation that may fail in edge cases and do not old in general
- simple, but well-designed experiments provide a sufficient empirical demonstration of the theoretical results

**Q4 Main Weakness:**

- Fig 3 & 5 do not show multiple runs (only single seed), the validity of the qualitative discussion in the running text (section 4 & 5) may be limited
- the outlined informal diagnostic does not seem of great help in high-dimensional settings to the reviewer

**Q5 Detailed Comments To The Authors:**

Thank you very much for your submission, indeed, very interesting work and superb clarity of the presentation.

Figures:
- 3 & 6: add label for the x-axis (assumed number of iterations and not epochs)
- 4: Indicate that variants of the A-VI are tested against a F-VI baseline, y-axis just reads VI algorithms
- 6: i=1 and i=2 in legend? indictate that the width k of the inference network
 for A-VI is varied

Questions:
- 3: what are the necessary requirements for the factorization according to a graph (theorem 3.4), i.e. which general class of graphical models is admissibale (DAGs, Markov fields, mixed graphs)

Typos:
- 1:
  - ",however**;**"
  - "posteriori dependence on (?)"
- 3:
  - Def. 3.2; comma after end of (i)
  - End Eq.10 with an period
  - Eq. 12 misses proper verb to complete the sentences or introduces the equation. The subsequent sentence could also be used to introduce Eq. 12 afterwards
- 4.4:
  - "a sin**g**le"
 - "faster than **F**_VI"?
- A2:
  - "**Theorem** 3.4"

Spacing:
- consider writing Eq. in two lines to avoid the column overflow
- consider placing Fig. 6 right to Fig.  5 and below of Fig. 4 on page 5 s.t. the discussion starts at page 8, or placing both on top on page 8

References: [Giordano et al., 24] has been published in JMLR

**Q9 Complying With Reviewing Instructions:**

Yes

---

> ### Author Rebuttal · Authors · 2024-04-03
>
> We thank the reviewer for their detailed reading of the paper, and for raising many interesting questions (and for their enthusiasm!).
>
> **Fig 3 and 5 do not show multiple runs.** We appreciate the limitation of these figures, and will further emphasize this point in the caption. The primary goal of the figures is to illustrate that sufficiently expressive inference networks allow A-VI to close the amortization gap. (We felt that putting multiple seeds would make the figure too cluttered, but we are open to suggestions.)
>
> To see the sensitivity to the random seed, Figure 4 presents results across all seeds, with an emphasis on time to convergence.
>
> **Outlined informal diagnostic.** We agree that running a few steps of F-VI after A-VI converges may not be enough to detect whether the inference function is sufficiently expressive. Indeed, it can take many iterations for the optimizer to resume moving for high-dimensional models, and more generally non-convex optimization landscapes. Per the reviewer’s suggestion, we will add this caveat to the discussion section.
>
> **Necessary requirements for class of graphical models.** Thank you for this question. Our reasoning is based on DAG representations, a point we can clarify when defining exchangeable latent variable models (Definition 3.2). It would be interesting to develop analogs to Theorem 3.4 for other classes of graphical models (Markov fields and mixed).
>
> We thank the reviewer for their attention to details and will update the paper to fix figures, typos and spacing, as recommended.

---

### Official Review · Reviewer_wGdd · 2024-03-21

**Q2-1 Originality-Novelty:** 2
**Q2-2 Correctness-Technical Quality:** 3
**Q2-5 Clarity Of Writing:** 3

**Q1 Summary And Contributions:**

Paper describes conditions based on CAVI rule in a latent variable model which when satisfied, amortized variational inference (A-VI) can be at par with the optimal solution yielded by factorized variational inference (F-VI). The authors focus on full Bayesian inference, instead of inferring only z. The authors mention three broad classes of models where, 1) conditions are satisfied (simple hierarchical models) and the amortized gap can be closed; 2) the conditions are not satisfied, but the amortization gap can be closed when the input domain to A-VI is expanded, 3) when amortized gap cannot be closed even after domain expansion. The authors provide some experiments with some latent variable models like Bayesian Neural Networks, saw Time series model, and si mple hierarchical models to prove the claim.

**Q2-3 Extent To Which Claims Are Supported By Evidence:**

2: Fair: the main claims are somewhat supported by evidence (but the experimental evaluation may be weak, or does not match entirely with the claims, important baselines may be missing, proofs contain important ideas but lack rigor, algorithmic details are only discussed superficially, references are imprecise, assumptions are not sufficiently motivated or explicated, etc.).

**Q2-4 Reproducibility:**

1: Poor: key details (e.g. proof sketches, experimental setup) are incomplete/unclear, or key resources (e.g. proofs, code, data) are unavailable.

**Q3 Main Strengths:**

The paper is organized well and is comprehensible.

The central idea of the paper revolving around the existence of an ideal inference function across different latent variable models is quite interesting, but I have not checked the proofs in detail. In section 3, wherever possible, the authors give an intuition to the readers of why an inference function may or may not exist. To my knowledge, this might be the first paper to analyze inference capabilities across different latent variable models following a principled way (incorporating CAVI rule). This means that for many dense hierarchical generative models that we see, it might be a good idea to do variational inference?

**Q4 Main Weakness:**

The experiments might be lacking some details -

1. Across experiments, what was the prior for \theta and z used?

2. For the experiments section, authors should be a bit more clear on the amortized distribution and it's training. For example, in section 4.3 and 4.4 does f_\phi output the parameters of a Gaussian distribution for both \theta and z? And then each optimization step updates only the parameters of the neural network f_\phi?

3. I understand either simulated data or FashionMNIST has been used for most experiments, why did the authors not consider more complex datasets and potentially deeper models for inference?

**Q5 Detailed Comments To The Authors:**

The authors mention in the end of the paper how the variational parameters can be optimized using (F-VI) after an initial set of approximations by A-VI. I wanted ask how semi-amortized inference fits into the analysis of closing the amortization gap? We know from some works that semi-amortized VI can help improve estimates to the posteriors after talking multiple optimization steps.

Semi-Amortized Variational Autoencoders, Kim et al, ICML 2018
Iterative Amortized Inference, Marino et al, ICML 2018

**Q9 Complying With Reviewing Instructions:**

Yes

---

> ### Author Rebuttal · Authors · 2024-04-03
>
> We thank the reviewer for their careful reading of the paper and their constructive feedback.
>
> **Priors in experiments.** For the linear model, we use the priors specified in equation 8 (flat for $\theta$ and standard normal for ${\bf z}$; note the flat prior on $\theta$ is chosen to simplify the calculations in Appendix A.4 and could be replaced with a normal prior). For all other experiments, we use a standard normal prior on both $\theta$ and ${\bf z}$. We will add these important details to Section 4, when defining the model.
>
> **Clarification on amortized distribution in experiments.** Thank you for your comment. We will clarify the details of the experiment as follows:
>
> *For all experiments, we use a factorized Gaussian approximation.
> We only use amortization of $\bf z$, given that, in a latent variable model, each $z_n$ corresponds to an observation $x_n$. This correspondence does not exist for $\theta$ and so we learn a separate Gaussian factor for each element of $\theta$ (see also the first paragraph in Section 2).
> The inference function maps $x_n$ to the mean and variance of the Gaussian factor for $z_n$. For the saw time series (Section 4.5) the input may also include $x_{n - 1}$.
> Each optimization step updates the parameters of the inference function (coefficients of the inference polynomial in 4.2, and weights of the inference network in Sections 4.3, 4.4, 4.5), as well as the variational parameters for $q(\theta)$.*
>
> We also emphasize that the code to reproduce our experiment will be available online as notebooks.
>
> **More complex data sets.** We appreciate the importance of studying more difficult problems. In this paper, we struck a balance between looking at illustrative examples and running experiments in a relatively controlled setting. In particular, we wanted some confidence that the optimizer converged to an optimum, since our theoretical results concern minimizers of the $\text{KL}(q||p)$. We spent our computation budget on trying a large number of inference functions, running many iterations, and repeating each experiment 10 times.
>
> In the discussion, we’ll highlight that further empirical studies–notably on problems where convergence cannot be achieved within a reasonable computational budget–as an important direction for future studies.
>
> **How does semi-amortized VI help?** We will add further details in our paper to clarify the role of semi-amortized VI. If A-VI convergences quickly to a suboptimal solution, it then makes sense to switch to F-VI to continue improving the solution. Starting with A-VI can jump start the optimization.
>
> We appreciated the references and will include Marion et al (2018) is the related work section. Kim et al (2018) is already cited.

---

### Official Review · Reviewer_sTqR · 2024-03-22

**Q2-1 Originality-Novelty:** 3
**Q2-2 Correctness-Technical Quality:** 3
**Q2-5 Clarity Of Writing:** 3

**Q1 Summary And Contributions:**

This paper studies amortized variational inference (which includes VAE). It provides (if and only if) conditions for AVI to achieve the same solution as MFVI. Basically, the latent variable model has to be a simple hierarchical model. Also, AVI is appropriate for full Bayesian inference, in the sense that the target distribution of p(theta, z | x).

**Q2-3 Extent To Which Claims Are Supported By Evidence:**

3: Good: the main claims are supported by convincing evidence (in the form of adequate experimental evaluation, proofs, (pseudo-)code, references, assumptions).

**Q2-4 Reproducibility:**

4: Excellent: key resources (e.g. proofs, code, data) are available and key details (e.g. proof sketches, experimental setup) are comprehensively described for competent researchers to confidently and easily reproduce the main results.

**Q3 Main Strengths:**

1. Theorem 3.4 gives sufficient and necessary conditions for the existence of an ideal inference function.

2. Providing a non-example in Proposition 3.8 is helpful for us to understand when the amortization gap cannot be closed.

3. The final paragraph on Page 5 provides some intuitions on why AVI works well on simple hierarchical models but not on HMM.

**Q4 Main Weakness:**

1. I think the writing and word choices can be improved in this paper. For example, "the modus operandi of VI" and "AVI is used as a cog". I do not believe "cog" and "modus operandi" are common phrases in academic use.

2. The introduction on AVI could be more accessible/self-contained for nonexperts in the field. For example, this article does not explain the motivation for using the inference function or why Q_A(F) is a poorer family than Q_F.

3. The experiments in this paper mainly concern the "number of interactions." The authors should provide comparisons on run time since each iteration of AVI has more cost than MFVI. It is unfair to say, "We also find that when the class of inference functions is sufficiently expressive, AVI often converges faster than F-VI to the optimal solution." when AVI costs more. This comparison is also unfair for AVI methods with different hidden layer widths k.

4. It is concerning that AVI is sensitive to random seeds in some cases, and I think this is worth discussing or mentioning in the introductions. The current summary paragraph for Section 4 on Page 2 is too strong, considering some of the experimental results I see. By the way, I do not think running each experiment 10 times supports the claims on "stability" for Task 4.4.

**Q5 Detailed Comments To The Authors:**

1. Page 4 has the sentence: "We show item (2) by starting with the CAVI rule for a". I need help finding where "item (2)" is.

2. On Page 3, there is a sentence "This strategy changes the factorization of q(θ, z) and is aimed at closing the inference gap, rather than the amortization gap." I get the idea that inference gap is hinting towards "ELBO", but I wish the author had been more specific about what the inference gap is.

3. On Page 6, the author mentions the parametrization trick, which should be from Kingma 2014. They should provide some reference.

**Q9 Complying With Reviewing Instructions:**

Yes

---

> ### Author Rebuttal · Authors · 2024-04-03
>
> We thank the reviewer for their detailed reading of the paper and their constructive comments.
>
> **Writing and choice of words.** As suggested, we can rephrase sentences which involve the words “cog” and “modus operandi”.
>
> **Introduction for A-VI.** At the top of page 2, we propose to add the following paragraph:
>
> *There exist several motivations for A-VI. One of them is scaling. While F-VI requires fitting a separate variational factor $q_n$ for each of the $N$ data points, A-VI can be more efficient since what we learn about $\phi$ can be amortized across data points. However, if A-VI’s inference function is not sufficiently expressive, it may fail to produce as sophisticated a solution as F-VI. We will formalize this intuition, and in fact go a little beyond, showing that no matter how expressive the inference function, $\mathcal Q_\text{A}$ is always a poorer family than $\mathcal Q_\text{F}$.*
>
> **Varying runtimes per iteration.** We thank the reviewer for this important suggestion and agree the wall time is an important metric to report. We generated new plots for Figures 3, 4, 5, and 6 which plot the wall time rather than the number of iterations. The plots can be seen via the (anonymized) link: https://docs.google.com/document/d/e/2PACX-1vT2tsZL-CskEDb0yQyvpKLi3Z3J5ZaGpCFFtNOy79bozsWuhcVNmwE8JkXjxSgLqoMe6UMq0eo_oT3T/pub
>
> For the linear, nonlinear and saw time-series, we find the number of iterations to be a good proxy for runtime, likely because the inference network is relatively small. For the Bayesian neural network, the cost per iteration for A-VI with a large network is noticeably larger than for F-VI. As a result, the speedup from using A-VI is more modest (25% speedup on average) and the wall time to convergence is the same for $k=128$ and $k=256$, even if the latter requires less iterations to converge. The reason, as you pointed out, is that the inference network with $k=256$ is more expensive to use.
>
> We will:
> * Replace all figures in Section 4 with the figures which plot wall time.
> * Include the current Figure 4 (against iterations) in the Appendix, and mention the trade-off between speed of convergence in terms of iteration and cost per iteration in Section 4.4.
>
> **Strength of statement in introduction.** We agree with the reviewer about the cautionary nature of some of our experiments. We propose to add a sentence at the end of the last paragraph in the **Plan** subsection:
>
> *We also find that when the class of inference functions is sufficiently expressive, A-VI often converges faster than F-VI to the optimal solution. However, in some problems, the performance of A-VI is sensitive to the random seed, potentially much more so than F-VI.*
>
> And in Section 4.3 (nonlinear probabilistic model, where A-VI is most sensitive to initialization):
>
> *However, A-VI is more sensitive to the seed and in some cases, can fail to converge after 20,000 iterations. A strength of F-VI relative to A-VI is therefore robustness to initialization, particularly in this example.*
>
> **Stability of Bayesian neural network.** Thank you for this comment. We have revised the statement in 4.4:
>
> *For $k \ge 64$, A-VI is consistently faster than A-VI across the 10 seeds used in our experiment.*
>
> **Item (2).** We will clarify that item (2) is the converse in Theorem 3.4, which states that if an inference function exists, the graphical model must be a simple hierarchical model.
>
> **What is the inference gap?** We will clarify in the **Related work** subsection that closing the inference gap means shrinking $\text{KL}(q||p)$ to 0, or equivalently further increasing the ELBO.
>
> Parameterization trick. We will add the suggested reference to Kingma 2014.

---

### Official Review · Reviewer_hW6L · 2024-03-22

**Q2-1 Originality-Novelty:** 3
**Q2-2 Correctness-Technical Quality:** 3
**Q2-5 Clarity Of Writing:** 4

**Q1 Summary And Contributions:**

This paper examines Amortized Variational Inference (A-VI) as a general-purpose method for approximate Bayesian inference, beyond its conventional use in training variational autoencoders. The authors provide a theoretical analysis to identify conditions under which A-VI can match the performance of Factorized Variational Inference (F-VI), thereby eliminating the so-called amortization gap. They establish that simple hierarchical models uniquely satisfy these conditions. Moreover, the paper discusses extending the domain of A-VI's inference function for broader model classes and highlights scenarios where closing the amortization gap is inherently infeasible.

**Q2-3 Extent To Which Claims Are Supported By Evidence:**

3: Good: the main claims are supported by convincing evidence (in the form of adequate experimental evaluation, proofs, (pseudo-)code, references, assumptions).

**Q2-4 Reproducibility:**

3: Good: key resources (e.g. proofs, code, data) are available and key details (e.g. proofs, experimental setup) are sufficiently well-described for competent researchers to confidently reproduce the main results.

**Q3 Main Strengths:**

- The paper contributes valuable theoretical insights by delineating the conditions under which A-VI can attain the optimal solution of F-VI, enhancing the general understanding of A-VI's applicability and limitations. By identifying model classes where A-VI is equally effective as F-VI, this work provides a clear guideline for practitioners on when to prefer A-VI, potentially simplifying inference in complex Bayesian models.

- The empirical evaluation spans various models, including linear and nonlinear probabilistic models, a Bayesian neural network, and time series, demonstrating the theory's applicability and exploring the practical aspects of implementing A-VI.

**Q4 Main Weakness:**

- (Minor): While the paper covers a significant range of models, which I consider sufficient for justifying the claims, the exploration of models where A-VI cannot close the amortization gap could be expanded further to include more diverse and complex examples.

**Q5 Detailed Comments To The Authors:**

The paper is well organized and very well written. The theoretical foundation is compelling and well-supported by rigorous proofs. Figure 1 is extremely helpful for understanding the findings.

After reading the paper, my major concern is:
- I would appreciate the authors elaborated more on Proposition 2.1 by explaining why $\nu_n  = \nu_m$ does not necessarily hold when $x_n = \x_m$.

**Q9 Complying With Reviewing Instructions:**

Yes

---

> ### Author Rebuttal · Authors · 2024-04-03
>
> We thank the reviewer for their comments.
>
> **More diverse experiments.** We understand the benefits of studying further examples from the deep learning and Bayesian literature, and also cases where convergence is not feasible within a reasonable compute time. (The latter is beyond the scope of our current theory which focuses on optimal solutions.) We see this as a fruitful avenue for future research.
>
> **$\nu_1 \neq \nu_2$ even when $x_1 = x_2$.** To show the strictness of the ordering in Proposition 2.1, it suffices to find an element of $\mathcal Q_\text{F}$ which does not belong to $\mathcal Q_\text{A}$. Note that this member need not be a minimizer of $\text{KL}(q||p)$. So if $x_1 = x_2$, we can simply construct such an element by setting $\nu_1 \neq \nu_2$. For example, $\mathcal Q_\text{F}$ may be the family of factorized Gaussians, and the means can be set to arbitrary (but distinct) values, even though $x_1 = x_2$.
>
> We will clarify this in the statement of the proof.
>
> Naturally, the more interesting problem is whether the $argmin_{q \in \mathcal Q_\text{F}} \in \mathcal Q_\text{A}$, and this is the main question we tackle in the remainder of the paper. For an explicit counter-example, where $x_1 = x_2$, but the optimal solution in $\mathcal Q_\text{F}$ verifies $\nu_1 \neq \nu_2$, see the HMM analysis in Appendix A.5, used to prove Proposition 3.8.

---

### Meta-Review · Area_Chair_gtnk · 2024-04-17

The reviewers tend to agree that the theoretical contributions of this work are significant despite a more limited experimental evaluation, and I tend to concur with this evaluation.  I believe that the reviewers have given good suggestions of how to improve the experiments, though I do agree with the majority of the reviewers that the theoretical results are enough to offset this limitation.  I do suggest that the authors take note of the most critical comments and try to improve the reproducibility of the experiments.